# The computational and learning benefits of Daleian neural networks

**Adam Haber**
Department of Brain Sciences
Weizmann Institute of Science
Rehovot, Israel
adam.haber@weizmann.ac.il

**Elad Schneidman**
Department of Brain Sciences
Weizmann Institute of Science
Rehovot, Israel
elad.schneidman@weizmann.ac.il

## Abstract

Dale's principle implies that biological neural networks are composed of neurons that are either excitatory or inhibitory. While the number of possible architectures of such Daleian networks is exponentially smaller than non-Daleian ones, the computational and functional implications of using Daleian networks by the brain are mostly unknown. Here, we use models of recurrent spiking neural networks and rate-based networks to show, surprisingly, that despite the structural limitations on Daleian networks, they can approximate the computation performed by non-Daleian networks to a very high degree of accuracy. Moreover, we find that Daleian networks are more functionally robust to synaptic noise. We then show that unlike non-Daleian networks, Daleian ones can learn efficiently by tuning single neuron features, nearly as well as learning by tuning individual synaptic weights – suggesting a simpler and more biologically plausible learning mechanism. We thus suggest that in addition to architectural simplicity, Dale's principle confers computational and learning benefits for biological networks, and offers new directions for constructing and training biologically-inspired artificial neural networks.

## 1 Introduction

The synaptic connectivity of neurons in the brain is shaped by a variety of physical, chemical, and biological factors [1, 2], even before learning comes into play. For example, the location of neurons and their morphology induce distance-dependent connectivity [3], whereas molecular lock-and-key mechanisms induce cell-type specific connectivity patterns [4]. The functional and computational implications of such constraints - from the architecture of connectomes to cell types - are fundamental to our understanding how real neural circuits are built and function.

A particularly stringent constraint on the design of biological neural networks is imposed by Dale's principle, which asserts that the axonal branches of a neuron release the same set of neurotransmitter molecules [5, 6]. While some exceptions to Dale's principle have been reported [7–10], the implied identity of neurons as either excitatory or inhibitory has been central to our characterization and understanding of neural circuits' structure and function [11–14]. Moreover, the balance between excitatory and inhibitory populations has been at the core of many computational models of the design of neural circuits and their function [15].

Dale's principle has far-reaching implications for the architecture of neural circuits: For directed networks of $N$ neurons, there are $N \cdot (N-1)$ possible synapses, and so for the general case in which each synapse can be either excitatory, inhibitory, or non-existent - there are $3^{N \cdot (N-1)}$ possible signed networks connecting the neurons. However, only $\left(2 \cdot 2^{N-1} - 1\right)^N$ of these networks abide by Dale's principle (see Supplementary Information, SI). Thus, already for 15 neurons, non-Daleian networks outnumber the Daleian ones by a factor of $\sim 10^{32}$. The staggering difference in richness of synaptic

36th Conference on Neural Information Processing Systems (NeurIPS 2022).

connectivity maps that non-Daleian networks offer compared to Daleian ones, and their potentially higher plasticity (the ability to change synaptic signs in learning) – suggest that non-Daleian networks may be able to carry computations that Daleian networks cannot. However, Daleian networks are clearly simpler in terms of biological design and development, since the set of neurotransmitters and receptors expressed by each neuron is smaller, compared to the expected requirements of regulation of connections in non-Daleian ones. Moreover, since synapses cannot change their sign during learning in Daleian networks, they are limited to a significantly smaller "search space", which might limit learning but could make it easier for Daleian architectures to find networks that perform a desired function. The obvious question is then: what is the brain "missing" by imposing Dale's principle on biological neural networks – if at all?

Analysis of the statistical properties of Daleian connectivity matrices in terms of their eigenvalue spectra suggested that the variance of synaptic weights in this case shapes network dynamics [16], and described the covariance structure of neural activity in such networks [17, 18]. One functional analysis of networks violating Dale's principle [19] suggested they may allow for robust balanced network states using fewer neurons than Daleian networks. Studies of training neural networks that obey Dale's principle suggested that feed-forward artificial neural networks (ANNs) with separate inhibitory and excitatory units can reach similar performance as feed-forward multi-layered perceptrons that are not sign-specific [20]. For spiking neural network models, a scheme for constructing networks that are compatible with biophysical constraints on connectivity and perform tasks that are solved by biological neural circuits, has been suggested in [21].

Here, we study the implications of Dale's principle on the function of recurrent neural networks, and their ability to learn. We simulate the responses of large ensembles of Daleian and non-Daleian networks to rich sets of stimuli, and measure the functional similarity between them in terms of the overlap of the distributions of their spiking responses. Surprisingly, we find that almost all non-Daleian networks have a Daleian "neighbor" that is close in functional space, namely it responds in a similar way to the same stimulus. Thus, the computations implemented by non-Daleian networks can be accurately approximated by Daleian ones. We analyze the sensitivity of networks to synaptic perturbations, and find that computations performed by Daleian networks are more robust to random synaptic noise. We further show that Daleian networks can learn to approximate the computation that arbitrary non-Daleian networks perform using a simpler and more biologically-plausible learning mechanism. Overall, our results suggest that Daleian networks are beneficial both from both a design perspective and a computational one, and offer potential venues for exploring biologically-inspired design of artificial neural networks.

## 2 Results

To characterize the functional differences between Daleian and non-Daleian networks, we studied the encoding of stimuli by these two kinds of networks, using two classes of models of recurrent neural networks: networks of spiking neurons and networks of firing rate-based neurons. We compared the responses of Daleian networks and non-Daleian ones for the same stimulus or class of stimuli, and explored the nature and implications of their differences.

### 2.1 Daleian and non-Daleian networks tile the space of neural response distributions in a similar manner

We started by considering the case of networks of $N$ spiking neurons, where the synaptic connectivity of a network is given by an $N \times N$ real matrix, $W$, where $W_{ij}$ is the synapse from neuron $i$ to neuron $j$, and $W_{ii} = 0$ for all $i$ (i.e., no self-synapses). The stimulus to the network is given by a vector $\mathbf{s} \in R^N$, such that $s_i$ is a time-independent stimulus to neuron $i$. The biases of neurons towards spiking are represented as $\boldsymbol{\theta} \in R^N$, such that $\theta_i$ sets the baseline activity level of neuron $i$. The activity of the network is discretized into time bins, such that a binary vector $\mathbf{x} \in \{0, 1\}^N$, denotes the spiking and silence of the neurons in time bin $t$, where $x_i = 1$ if neuron $i$ spiked in that time bin, and $x_i = 0$ otherwise; time bins were set here to $20\,ms$. The dynamics of the network is given by

$$P_{W,\boldsymbol{\theta}}\left(\mathbf{x}_t | \mathbf{x}_{t-1}, \mathbf{s}\right) = \frac{1}{Z} \exp\left(\mathbf{x}_{t-1}^T \cdot W \cdot \mathbf{x}_t + \boldsymbol{\theta} \cdot \mathbf{x}_t + \mathbf{s} \cdot \mathbf{x}_t\right) \tag{1}$$

so the probability of neuron $j$ to spike in time $t$ is given by a sigmoidal function of the sum of its synaptic inputs $\sum_i W_{ij} \cdot x^i_{t-1}$, its bias towards spiking is $\theta_j$, the stimulus it receives is $s_j$, and $Z$ is the partition function. Equation (1) defines a transition matrix between the $2^N$ binary states of the network, $M$, whose entries are all strictly positive (due to the definition of eq. 1), and so this Markov chain converges to a unique stationary distribution over the states of the network for a given stimulus $\pi_{W,\theta}(\mathbf{x}|\mathbf{s})$. This unique stationary distribution satisfies $\pi_{W,\theta}(\mathbf{x}|\mathbf{s}) \cdot M = \pi_{W,\theta}(\mathbf{x}|\mathbf{s})$, or $\pi_{W,\theta}(\mathbf{x}|\mathbf{s}) \cdot (M - I) = 0$, where $I$ is the identity matrix. Since $\pi_{W,\theta}(\mathbf{x}|\mathbf{s})$ is a vector from the left nullspace of $(M - I)$, and we know that this nullspace has dimension 1, $\pi_{W,\theta}(\mathbf{x}|\mathbf{s})$ can be uniquely found by singular value decomposition (SVD) of $(M - I)$ (see Methods).

To compare Daleian and non-Daleian networks, we computed the stationary distributions $\pi_{W,\theta}(\mathbf{x}|\mathbf{s})$ of 5000 Daleian networks and 5000 non-Daleian networks of $N = 10$ neurons, each presented with 100 randomly selected stimuli (Fig. 1a-c). For the non-Daleian (nD) networks, synapses were sampled from a Gaussian distribution $W^{\mathrm{nD}}_{ij} \sim \mathcal{N}(0, \frac{1}{\sqrt{N}})$, where $i, j \in 1 \ldots N$. For the Daleian (D) networks, half of the neurons were selected to be excitatory and half inhibitory, and the outgoing synaptic weights of each neuron were sampled from the positive or negative parts of a normal distribution, namely, $W^{\mathrm{D}}_{ij} \sim |\mathcal{N}(0, \frac{1}{\sqrt{N}})| \cdot \psi(i)$, where $\psi(i) = 1$ for excitatory neurons and $-1$ for inhibitory ones. The stimuli were sampled from a normal distribution, $s_i \sim \mathcal{N}(0, 1)$ ($i \in 1 \ldots N$), as in this range the network responses were not dominated by the stimulus or the recurrent activity in the network, but by the combination of both (see SI). Importantly, while the stationary distributions are an asymptotic property of the model, the rate of convergence of these Markov chains for our networks is very fast, approaching $\pi_{W,\theta}(\mathbf{x}|\mathbf{s})$ at an exponential rate (the rate is governed by the spectral gap of $M$, which here corresponds to about 5 time steps, or $100\ ms$; see SI). We note that the dynamics defined by eq. (1) converge to $\pi_{W,\theta}(\mathbf{x}|\mathbf{s})$ *irrespective* of the initial conditions of the network, which justifies the comparison of networks by the stationary distributions of their responses to the same stimulus.

We quantified the functional similarity between all pairs of networks within and between the two ensembles, using the dissimilarity of the stationary response distributions:

$$D_{func}(W_k, W_l | \mathbf{s}) = D_{JS}\left[\pi_{W_k,\theta_k}(\mathbf{x}|\mathbf{s}) || \pi_{W_l,\theta_l}(\mathbf{x}|\mathbf{s})\right]. \tag{2}$$

where $D_{JS}$ is the Jensen-Shannon divergence – a bounded and symmetric measure of the overlap of the distributions [22] (see Methods). Figure 1d shows a sub-sample of the resulting $10,000 \times 10,000$ dissimilarity matrix between all networks in the Daleian and non-Daleian ensembles for one such stimulus. The dissimilarity matrices for different stimuli were highly correlated (see SI). The structure of the dissimilarity matrix suggests that Daleian networks do not inhabit a specific part of functional space, but rather that their response distributions tile the space of stationary distributions in a similar way to that of the non-Daleian networks. This is further demonstrated by the 2-dimensional embedding of the dissimilarity matrix (Fig. 1e), which does not show any obvious separation between Daleian and non-Daleian networks. Furthermore, the profiles of distances from a randomly chosen Daleian or non-Daleian network to the other networks are consistent over different choices of networks, as shown in Fig. 1f,g - reflecting that for a typical non-Daleian network, there is a Daleian network which is functionally close to it. We further computed for each stimulus the percent of non-Daleian networks whose nearest network over the two ensembles was a Daleian network, and found that, across all stimuli, the probability that the nearest network to a non-Daleian network would be Daleian is approximately 50%, reflecting, again, that Daleian networks cover the space of response distributions in a similar manner to that of non-Daleian ones. Importantly, repeating the analysis for the relations between networks, averaged over the set of stimuli, namely the matrix given by $\langle D_{func}(W_k, W_l | \mathbf{s}) \rangle_{\mathbf{s}}$ (where $\langle \rangle_{\mathbf{s}}$ denotes averaging over all stimuli) gave similar structure in terms of the "coverage" of the same space (see SI).

Thus, the results for our sampled networks and stimuli suggest that for a given non-Daleian network, there is a Daleian network that is nearby in terms of its function. To verify that this is not the result of the details of the spiking neural network models that we used, our sampling of the space of non-Daleian and Daleian networks, or the size of the networks – we studied different distributions of synaptic weights (see SI), as well as another class of network models and larger networks' size (see below). Moreover, for both classes of network models, we asked, rather than relying on sampling the space of networks – how well may Daleian networks learn to approximate non-Daleian ones?

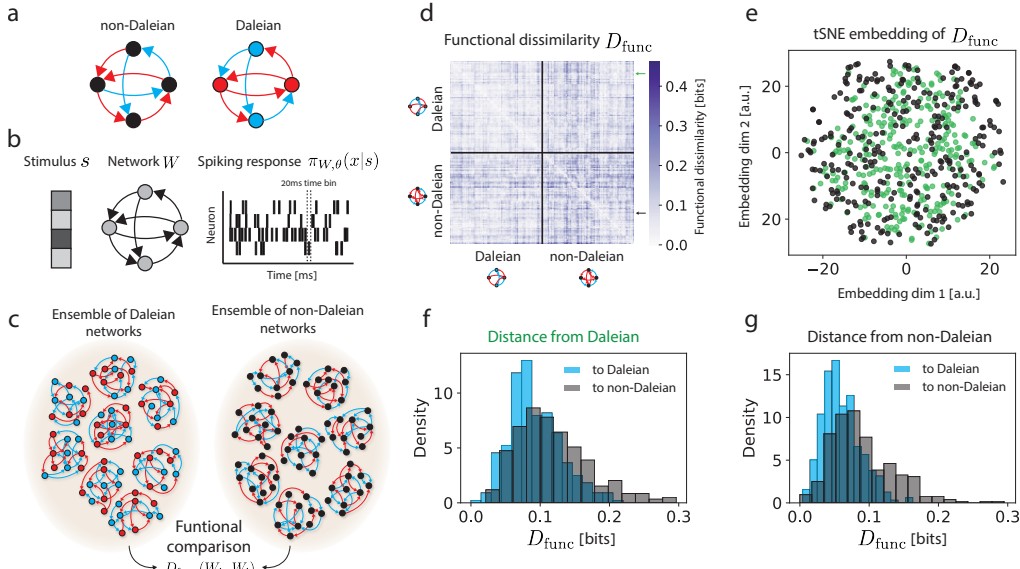

Figure 1: **Daleian and non-Daleian recurrent neural networks cover the same functional space.**
(**a**) In non-Daleian networks, the same neuron can have both excitatory and inhibitory outgoing synapses. In Daleian networks, all the synapses going out of a neuron are either exclusively excitatory (red neurons) or inhibitory (blue neurons). (**b**) We simulate the responses of networks of spiking neurons whose connectivity is given by $W$ to a large set of random stimuli, $\{\mathbf{s}_k\}$, and compute the stationary response distribution to each stimulus $\pi_{W,\theta}(\mathbf{x}|\mathbf{s}_k)$. (**c**) We sample 10,000 Daleian and non-Daleian networks, and compute the response distribution of all networks to the same stimuli. We then measure the functional dissimilarity between networks by the overlap of these distributions. (**d**) An example of the pairwise functional dissimilarity matrix between 300 Daleian and 300 non-Daleian networks presented with the same stimulus (only 600 networks are shown for visualization purpose). Rows and columns within the block of Daleian vs. Daleian networks (and non-Daleian vs. non-Daleian ones), were each ordered by hierarchical clustering (see Methods). The black horizontal and vertical black lines mark the transition from Dale to non-Dale networks. (**e**) A 2-dimensional tSNE embedding of the dissimilarity matrix from (d). Each dot marks one network, with Daleian networks shown in green, and non-Daleian shown in black. (**f**) A representative example of the distributions of functional dissimilarity values between one randomly chosen Daleian (marked by a green arrow in (d)) network and the rest of the networks in the ensemble. (**g**) Similar to (f) but for a randomly selected non-Daleian network (black arrow in (d)). Distance profiles shown in (f) and (g) are typical for all networks in the ensemble (SI).

## 2.2 Daleian networks learn to approximate non-Daleian ones with high accuracy

We asked how accurately we can approximate the response of a randomly selected non-Daleian network with synaptic connectivity $W^{nD}$ to stimulus $\mathbf{s}$, using a Daleian network $W^D$, by optimizing its synaptic weights. We started from a randomly chosen Daleian connectivity matrix $W^D_{\text{init}}$ with continuous synaptic weights, where half of the neurons were randomly chosen to have only positive outgoing synapses, the other half had only negative outgoing synapses, and the diagonal terms were all zero (Fig. 2a). We used gradient descent on $W_D$ to minimize $\mathcal{L}(W^D) = D_{JS}\left[\pi_{W^D,\boldsymbol{\theta}}(\mathbf{x}|\mathbf{s})||\pi_{W^{nD},\boldsymbol{\theta}}(\mathbf{x}|\mathbf{s})\right]$. To keep the excitatory/inhibitory identity of neurons, we optimized the *magnitude* of their synaptic weights, maintaining their sign throughout learning.

We performed such learning for 2000 different non-Daleian networks $W^{nD}$, each with a different stimulus $\mathbf{s}$, yielding 2000 Daleian approximations (see Methods for optimization details). We found that for a random non-Daleian network $A$ there exists, with high probability, a Daleian network $B$ that is orders-of-magnitude closer than a typical non-Daleian network $C$. Over the whole set, optimized Daleian networks were two orders-of-magnitude closer to target networks compared to the typical functional distance between networks in our sampled ensembles above, and one order-of-magnitude

closer than the nearest network in the ensemble (Fig. 2b). The average dissimilarity of an optimized Daleian network to a random non-Daleian responding to arbitrary stimuli reached $D_{func} \sim 5 \cdot 10^{-4}$ bits, which means that 2000 time bins are needed to distinguish between the responses of the target non-Daleian network and the optimized Daleian one. (For reference, optimizing a non-Daleian network to approximate a non-Daleian network gives $D_{func} \sim 3 \cdot 10^{-5}$ bits).

We then asked how well can an optimized Daleian network approximate a non-Daleian one on a set of stimuli, i.e. how similar they are in terms of the function they compute. We therefore repeated the learning described above, but this time minimizing $\langle \mathcal{L}(W^D) \rangle_{\mathbf{s}}$. Again, the optimized Daleian networks were highly accurate in approximating the non-Daleian ones, now reaching an average $D_{func}$ value of $\sim 6 \cdot 10^{-3}$ bits, when training only $W^D$ or when training both $W^D$ and $\theta$ (Fig. 2c). These differences mean it would take about 200 time bins to tell apart the responses of the Daleian network from the non-Daleian one; for time bins of $20\,ms$, this means it would take more than $\sim 4000\,ms$ of neural activity to the networks apart.

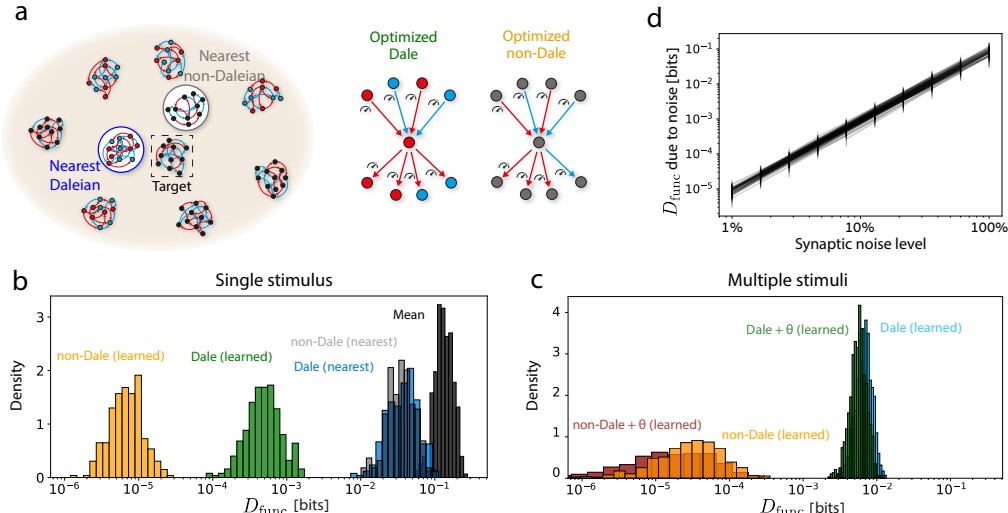

Figure 2: **Daleian networks can learn to accurately approximate the function of non-Daleian networks.** (**a**) For a given non-Daleian network $W^{nD}$ and stimulus $\mathbf{s}$, we compute the distance to the nearest Daleian network in the sampled ensemble (blue) and the nearest non-Daleian network in the sampled ensemble (gray). We also optimize a random Daleian network to approximate the target non-Daleian network (these learned networks will be later denoted in green) and learn a non-Daleian network that is optimized to approximate the non-Daleian network (later shown in orange). In the last two cases, learning is done by gradient descent on all synapses, see text. (**b**) The histograms of distances of the sampled and learned networks from 2000 different non-Daleian networks and 2000 different stimuli. Different colors show the distances to the target non-Daleian networks from the different classes of networks; color as described in (a). For reference, we also show a histogram of the mean distance from each non-Daleian network to all other networks in the ensemble for the corresponding stimulus (black). (**c**) The histograms of distances of the learned networks from 2000 different non-Daleian, each responding to 30 different stimuli. Different colors show the distances to the target non-Daleian networks from the different classes of networks (Daleian/non-Daleian), such that learning is done either by optimizing synaptic weights alone or both the synaptic weights and the neuronal thresholds $\boldsymbol{\theta}$. (**d**) Functional dissimilarity between the stationary distribution of a network $W^{nD}$ responding to stimulus $\mathbf{s}$, and noisy variations of $W^{nD}$ for different levels of synaptic noise (noise is independent for each synapse). Error bars correspond to standard deviations over 100 different noise realizations. Results shown for 30 different networks and 30 different stimuli.

We emphasize that we did not optimize the types of neurons of Daleian networks that learn to approximate a non-Daleian one, i.e. we did not pick which neurons would be excitatory and which inhibitory in each network. This means that our optimized Daleian networks are an upper bound on the ability of Daleian networks to approximate non-Daleian ones, and thus reflect how capable Daleian networks are. Interestingly, the learned networks had significantly broader distributions

of synaptic weights, and Daleian optimized networks were significantly sparser than non-Daleian optimized networks, similar to the properties of real neuronal circuits (see SI).

To give a more biologically relevant interpretation to these differences, we asked what level of synaptic noise would result in similar functional dissimilarity values. Thus, for randomly sampled network $W^{nD}$ and stimulus $\mathbf{s}$, we computed the Jensen-Shannon divergence between the stationary distribution of the original $W^{nD}$ and noisy versions of it, $W_\epsilon^{nD} = W^{nD} \odot \epsilon$, for different levels of multiplicative synaptic noise $\epsilon$ (see SI). Averaging over noise realizations, networks, and stimuli – we found that a functional dissimilarity value of $D_{JS}[\pi_{W^{nD},\boldsymbol{\theta}}(\mathbf{x}|\mathbf{s})||\pi_{W_\epsilon^{nD},\boldsymbol{\theta}}(\mathbf{x}|\mathbf{s})] \sim 5 \cdot 10^{-3}$ bits, a typical value for the optimized Daleian approximation, corresponds to synaptic noise level of $\sim 25\%$ (Fig. 2d). Given the variability of activation strength of synaptic connections in real neural networks, and their dynamic nature, these results suggest that Daleian networks can approximate non-Daleian ones to a degree that is very close to the limit set by synaptic and neuronal noise [23].

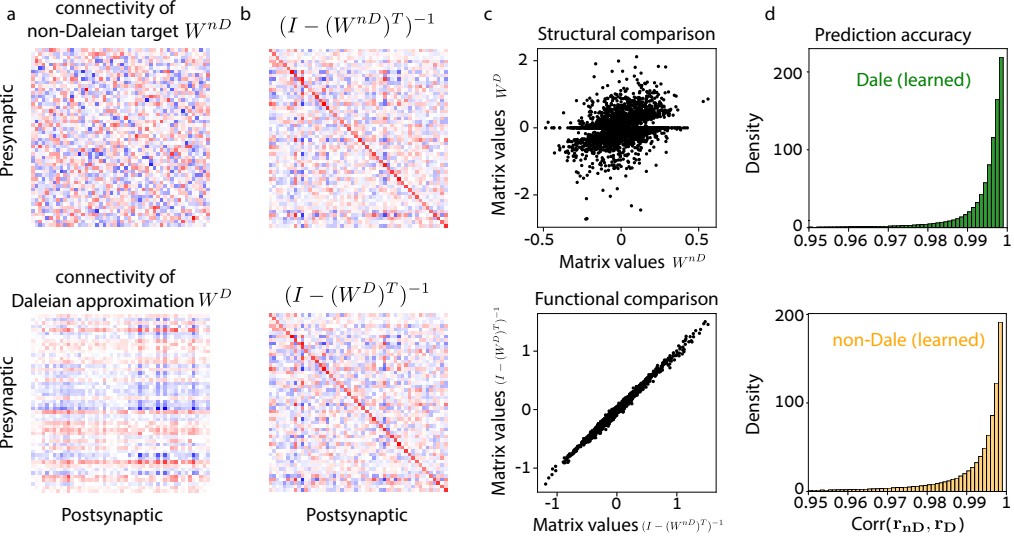

Figure 3: **Daleian rate-based networks can learn to accurately approximate rate-based non-Daleian networks** (**a**) An example of the connectivity matrix of a non-Daleian target the continuous-time recurrent firing-rate network of 50 neurons, $W^{nD}$ (top) and the connectivity of its Daleian approximation $W^D$. (**b**) Corresponding $\left(I - (W^{nD})^T\right)^{-1}$ matrix (top) and $\left(I - (W^D)^T\right)^{-1}$ matrix (bottom). (**c**) Comparison of individual entries in the non-Daleian network and its Daleian approximation: structural similarity is shown for connectivity matrices (top); functional comparison (bottom). (**d**) Accuracy of the Daleian approximation (green) and a non-Daleian approximation (orange) for firing-rate-based model networks with $N = 100$ neurons, measured by the correlation of their steady state activity vectors, averaged over 1000 different stimuli.

Next, we studied how well Daleian networks can approximate non-Daleian ones using another commonly studied class of network models: Continuous-time recurrent firing-rate networks [24], which offer mathematical tractability, and also allowed us to extend our analysis to much larger networks. Here, neural activity is described by firing rates of the $N$ neurons, which are denoted by $\mathbf{r} \in R^N$. Correspondingly, the dynamics is given by a differential equation with integration time constant $\tau$, and so the response of the network to a stimulus $\mathbf{s}$, is given by

$$\tau \frac{d\mathbf{r}}{dt} = -\mathbf{r} + W^T\mathbf{r} + \mathbf{s} , \tag{3}$$

and the steady state solution is given by $\mathbf{r}_{ss} = \left(I - W^T\right)^{-1}\mathbf{s}$. We asked how well we can approximate the computation performed by an arbitrary rate-based network $W^{nD}$ of $N = 100$ neurons using a Daleian network $W^D$. We repeated the analysis we performed for spiking networks, but here, for each randomly chosen non-Daleian network we used gradient descent to minimize the loss function:

$$\mathcal{L}(W^D) = \left\| \left(I - (W^{nD})^T\right)^{-1} - \left(I - (W^D)^T\right)^{-1} \right\|^2 .$$

We found that the optimized Daleian matrices $W^D$ are very different from $W^{nD}$ of the non-Daleian network they approximate (Fig. 3a,c), while $\left(I - (W^D)^T\right)^{-1}$ are similar to $\left(I - (W^{nD})^T\right)^{-1}$ (Fig. 3b,c). Thus, non-Daleian networks and their Daleian approximations have very similar stimulus-to-steady-state mapping: The mean Pearson correlation, for 1000 stimuli, between the steady states of non-Daleian targets and those of their Daleian approximations was $\sim 99\%$ (Fig. 3d). These correlation values were very similar to the accuracy of learning to approximate a non-Daleian networks by another non-Daleian network.

We conclude that non-Daleian networks can be approximated with high accuracy by Daleian ones. Moreover, since our learning of the optimized Daleian approximations is based on simple gradient descent and randomly chosen initial Daleian network structure, it is clear that our results are an upper-bound on how close one can get to a non-Daleian network with a Daleian one. Having established that there is no clear functional loss in using Daleian networks compared to the supposedly more powerful non-Daleian ones, we turned to explore the sensitivity and learning dynamics of Daleian and non-Daleian networks, and sought functional benefits of using Daleian networks.

### 2.3 Robustness and adaptability of Daleian networks are superior to non-Daleian ones

In learning and optimizing their function, neural networks need to balance changes that improve their performance while retaining their previously learned function. This suggests that certain network changes would result in altered function, whereas others should not. To explore the "design" of learnability of non-Daleian and Daleian networks and their robustness, we next asked how sensitive they are to changes in their synaptic connectivity maps and to the properties of individual neurons.

We first explored the effect of changes in synaptic weights on the function of a given network by computing $f_W\left(\Delta W\right) = D_{KL}[\pi_{W,\boldsymbol{\theta}}(\mathbf{x}|\mathbf{s})||\pi_{W+\Delta W,\boldsymbol{\theta}}(\mathbf{x}|\mathbf{s})]$ and then the Hessian of $f$, given by the matrix $[H_f]_{ij,kl}^W = \frac{\partial^2 f}{\partial W_{ij}\partial W_{kl}}$. We note that in this case, the Hessian of $H_f$ (evaluated at $W$) is the Fisher Information Matrix of $\pi_{W,\boldsymbol{\theta}}(\mathbf{x}|\mathbf{s})$, which provides a principled interpretation of the analysis and the results that follow [25]. The Hessian measures the curvature of $f$ in all directions of parameter space, where each direction corresponds to changing the weight of a single synapse. The trace of the $H$ then measures the overall "flatness" of $f$ around $W$. From a biological perspective, the flatness corresponds to the robustness of the network to synaptic fluctuations. We thus computed $\mathrm{Tr}\left[H_f\right] = \sum_{i,j\in 1\ldots N, i\neq j} \frac{\partial^2 f}{\partial W_{ij}^2}$, for each of the networks in the Daleian and non-Daleian ensembles, and for each sampled stimulus. To compare the sensitivities of Daleian and non-Daleian networks, we computed the ratio between the average trace of Daleian networks and that of non-Daleian networks, separately for each stimulus. This ratio quantifies whether Daleian networks are more sensitive (ratio $> 1$) or less sensitive (ratio $< 1$) to random synaptic changes compared to non-Daleian networks. We found that for the 98% of the stimuli, Daleian networks were *less* sensitive than non-Daleian ones to perturbations of synaptic weights (Fig. 4a). Given the inherent noise in biological synapses, such robustness carries a clear functional advantage.

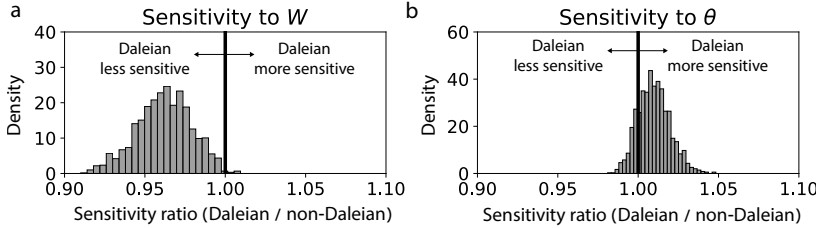

Figure 4: **Daleian networks are less sensitive to synaptic fluctuations and more sensitive to changes of single neuron parameters.** (**a**) Histogram of the ratio of the average sensitivity to synaptic perturbation of Daleian and the average sensitivity of non-Daleian networks, shown for 1000 different stimuli. Average sensitivity is measured using the trace of each of the Hessian matrices of 1000 networks for a given stimulus. (**b**) Similar to (a) but sensitivity is computed with respect to neuronal biases vector $\boldsymbol{\theta}$.

Increased robustness to synaptic fluctuations may hinder the ability of a network to adjust its reactions to changes in the environment. We therefore repeated the sensitivity analysis above, but this time computed the Hessian with respect to changes in the bias of individual neurons $\theta_i$, $[H_f]_{ij}^{\boldsymbol{\theta}} = \frac{\partial^2 f}{\partial \theta_i \partial \theta_j}$ and computed $\mathrm{Tr}\,[H_f] = \sum_{i \in 1...N} \frac{\partial^2 f}{\partial \theta_i^2}$, for each of the networks and each stimulus. We then computed the ratio between the average sensitivity of Daleian networks to threshold perturbations, and that of non-Daleian networks, separately for each stimulus. We found that for 84% of the stimuli, Daleian networks were, on average, *more* sensitive to changes in the threshold parameters compared to non-Daleian ones (Fig. 4b). Thus, the differences in sensitivities of the two classes of networks mean that Daleian networks are more resilient to random synaptic fluctuations, and may learn effectively by tuning the baseline firing rates of individual neurons.

## 2.4 Daleian networks learn efficiently by tuning only single neuron properties

We asked how learning by changing the properties of individual neurons, rather than individual synapses, would work for Daleian and non-Daleian networks. We evaluated learning performance using the loss function:

$$\mathcal{L}(W) = D_{JS}\left[\pi_{\text{target}}(\mathbf{x}) || \pi_{W,\boldsymbol{\theta}}(\mathbf{x}|\mathbf{s})\right]$$

where $\pi_{\text{target}}$ is the distribution that we wish to approximate with the optimized network, $W$. Given the homeostatic scaling of changes in synaptic weights [26] and the identification of "architectural" features that shape the function of neural networks [27, 28], it is clear that learning by adapting neuronal features could have computational benefits and be more plausible or efficient than learning by changing individual synapses. We thus compared the results of learning by gradient descent over $\{W_{ij}\}$, to learning by "macroscopic", neuron-level features. Specifically, we used a scaling factor for all outgoing synapses of a neuron $\lambda_i^{out} > 0$, a scaling factor for all incoming synapses of a neuron $\lambda_i^{in} > 0$, and the firing bias $\theta_i$: during learning, each synapse is re-scaled according to the pre-synaptic and post-synaptic scaling factors $W_{ij} \rightarrow \lambda_i^{out}\lambda_j^{in} \cdot W_{ij}$ ($\lambda_i^{out}$ and $\lambda_i^{in}$ are initialized to 1 before learning), and each neuron's bias is adapted individually $\theta_i \rightarrow \theta_i + \Delta\theta_i$ ($\Delta\theta_i$ is initialized to 0 before learning). Thus, instead of learning by changing the $N(N-1)$ terms of the full synaptic connectivity map, we learned by changing $3N$ parameters, corresponding to known biological features at the level of individual neurons (Fig. 5a). Clearly, synaptic learning is more potent than the neuron-level learning (any learning trajectory in weights-space due to neuronal learning is also achievable by synaptic learning, but not vice versa). But, it turns out that they act very differently in terms of the accuracy of synaptic and neuron-level learning on Daleian vs. non-Daleian networks.

We then asked how well the different classes of networks would approximate a distribution $\pi_{\text{target}}$ corresponding to a natural response distribution of biological neural networks. We used real spiking patterns of groups of 10 neurons recorded from the prefrontal cortex of macaque monkeys performing a visual classification task [29], to compare the accuracy of synaptic learning and neuron-level learning in mapping a random stimulus to the empirical response distributions of real neurons. We found that while both Daleian and non-Daleian networks achieved higher accuracy using synaptic learning (Fig. 5b), learning by changing single neurons' properties achieved higher performance for Daleian networks compared to non-Daleian networks (Fig. 5c,d). This suggests that the constraints set by Dale's principle change the relative effectiveness of these learning mechanisms.

## 2.5 Information coding by Daleian networks

As we have shown that Daleian networks are able to accurately approximate the responses of non-Daleian networks for a given stimulus, we asked how the two classes may differ in their encoding of a set of stimuli, which would reflect on the ability of downstream neurons to decode stimuli from network responses. We therefore computed the information that the responses of a network whose synaptic connectivity is given by $W$ carries about a set of $k = 500$ different stimuli. We first computed the entropy of the stationary distribution for each stimulus, $H\left[\pi_{W,\boldsymbol{\theta}}(\mathbf{x}|\mathbf{s}_i)\right] = -\sum_{\mathbf{x}} \pi_{W,\boldsymbol{\theta}}(\mathbf{x}|\mathbf{s}_i) \cdot \log_2 \pi_{W,\boldsymbol{\theta}}(\mathbf{x}|\mathbf{s}_i)$.

The total entropy of the network's activity is given by $H\left[\pi_{W,\boldsymbol{\theta}}\right] = -\sum_{\mathbf{x}} \pi_{W,\boldsymbol{\theta}}(\mathbf{x}) \cdot \log_2 \pi_{W,\boldsymbol{\theta}}(\mathbf{x})$, where $\pi_{W,\boldsymbol{\theta}}(\mathbf{x}) = \frac{1}{k}\sum_{i=1...k} \pi_{W,\boldsymbol{\theta}}(\mathbf{x}|\mathbf{s}_i)$, where we assumed a uniform distribution over stimuli (using non-uniform stimulus distributions gave similar results; see SI). We then computed the mutual

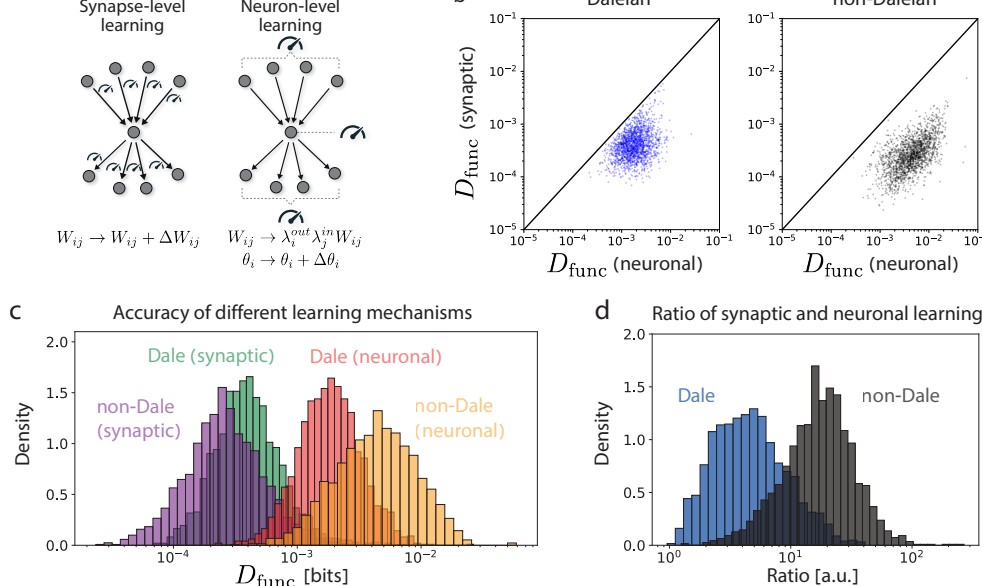

Figure 5: **Daleian networks can learn efficiently by tuning single-neuron properties.** (**a**) In synaptic learning each synapse is separately tuned (left), while in neuronal learning, changes to neuronal properties induce correlated changes across synapses of the changed neuron (right; see text). (**b**) Accuracy of synaptic learning (measured by functional similarity to $\pi_{\text{target}}$; lower $D_{func}$ means higher accuracy) is compared to neuron-level learning. Points show the results for 2000 different $(\mathbf{s}, \pi_{\text{target}})$ pairs; each point shows the mean $D_{func}$ when learning the mapping from a random stimulus $\mathbf{s}$ to a random target distribution $\pi_{\text{target}}$, over 30 Daleian (left) or non-Daleian (right) initial networks. (**c**) Comparison of the learning accuracy of Daleian and non-Daleian networks using synaptic and neuron-level learning mechanisms. Histograms show the distributions of the average $D_{func}$ over 30 Daleian/non-Daleian networks trained to map a random stimulus $\mathbf{s}$ to a target distribution $\pi_{\text{target}}$. (**d**) The ratio between the accuracy of synaptic learning and neuron-level learning for Daleian and non-Daleian networks, as measured by the ratio of mean $D_{func}$ values from (b).

information between the stimuli and the stationary responses,

$$I_W(\mathbf{s}; \mathbf{x}) = H[\pi_{W,\theta}(\mathbf{x})] - \langle H[\pi_{W,\theta}(\mathbf{x}|\mathbf{s})]\rangle_{\mathbf{s}} \tag{4}$$

where $\langle\rangle_{\mathbf{s}}$ denotes averaging over stimuli. For 500 Daleian networks and 500 non-Daleian networks of $N = 10$ neurons, we found that for randomly sampled networks, the responses of Daleian networks were more informative about the stimulus. Then, using both synaptic and neuronal learning, we trained the networks to maximize the mutual information in eq. (4) (Fig. 6a). We found that Daleian networks reach similar performance using synaptic learning, but are more informative (on average) when using neuron-level learning (Fig. 6b).

## 3    Discussion

We found that despite the clear constraint that Dale's principle imposes on the architecture of neural networks, it does not seem to incur a clear cost in terms of the computational capacity of Daleian networks or their ability to learn. We were able to approximate the responses of arbitrary non-Daleian networks to a range of stimuli, with Daleian networks, to a very high degree of accuracy. In fact, this accuracy is so high that it is on par with the level of intrinsic noise level of real biological networks. Moreover, we found that Daleian networks are more robust to synaptic noise, and that they can learn by tuning of neuron-level parameters, nearly as well as can be achieved by general synaptic learning rules. Our results suggest that in addition to the clear structural and developmental benefits of using Daleian neural networks, the brain may actually gain from using them, as they offer similar computational power, higher robustness to noise, and simpler learning mechanisms.

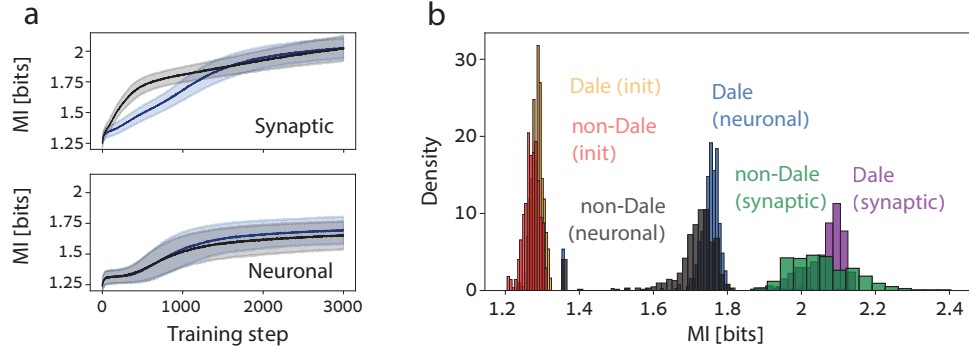

Figure 6: **Random Daleian networks are more informative than random non-Daleian ones, whereas trained Daleian networks carry similar information to trained non-Daleian.** (**a**) The mean of the information values carried by 500 Daleian (blue) and 500 non-Daleian (gray) networks, is shown as a function of training steps. Each network was trained to optimize its information on a given set of 500 stimuli using synaptic learning (top) or neuronal learning (bottom). Error bars represent 1 standard deviation. (**b**) Distribution of information values carried by networks on a set of 500 random stimuli before learning: Daleian networks shown in yellow, non-Daleian in orange. Distribution of information values of network trained to maximize the carried information by neuronal learning (Daleian in blue, non-Dale in gray), and using synaptic learning (Dale in purple, non-Dale in green).

Our analysis is naturally limited by the network models we have explored, our use of dense and relatively small networks, and the choices of connectivity patterns and stimulus statistics. Analysis of a wider range of connectivity parameters, log-normally distributed synaptic weights [30], and richer stimuli have given similar results, yet the exploration of larger and sparser networks, more naturalistic stimuli, and different connectivity patterns is warranted.

Other differences between Daleian and non-Daleian networks suggest future directions of study: Randomly sampled Daleian networks, as well as ones optimized to approximate non-Daleian ones show pairwise correlations profile that is skewed towards positive correlations, similar to real neural networks, whereas randomly sampled non-Daleian ones show balanced distribution of correlations (see SI). These differences suggest organization features of Daleian networks that remain to be uncovered. Somewhat diametrically, we note that real networks, which are mostly Daleian, have been shaped by evolution, development, and learning to implement specific computations. It is likely then that within the vast space of non-Daleian networks there would be ones that have interesting and distinct architectures both in terms of structure and the computations they carry.

The potential implications of our results for ANNs are also intriguing. Theoretical studies of Deep Neural Networks and their applications have usually ignored Dale's principle, as its inclusion typically impairs learning. Recent results on potential remedies that would allow ANNs to rely on distinct classes of neurons [31] and in particular Excitatory and Inhibitory ones [20] suggest how the integration of design features and properties of real neural networks into artificial ones might be beneficial [32, 33]. In particular, the idea that structural constraints on networks can affect learning trajectories and define the loss surface has a long history in constructing and training of artificial neural networks [34–36]. We hypothesize that some of the benefits that Dale's principles confers on recurrent biology-inspired neural networks may extend to artificial ones. It will be especially interesting to explore the parallels of our results in terms of shaping the search space of sign-constrained ANNs, the possibility of efficient learning by neuronal parameters rather than synaptic ones, the robustness and sensitivity they may offer, and their potential for preventing catastrophic forgetting.

## Acknowledgments and Disclosure of Funding

This work was supported by Simons Collaboration on the Global Brain grant 542997, Israel Science Foundation grant 137628, Israeli Council for Higher Education/Weizmann Data Science Research Center, Martin Kushner Schnur, and Mr. & Mrs. Lawrence Feis. ES is the incumbent of Joseph and Bessie Feinberg Chair.

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
