# OpenReview forum: "The computational and learning benefits of Daleian neural networks"
_NeurIPS.cc/2022/Conference — NeurIPS 2022 Accept_

### Official Review · Reviewer_eksJ · 2022-07-11

**Rating:** 8
**Confidence:** 5
**Soundness:** 3 good
**Presentation:** 4 excellent
**Contribution:** 3 good

**Summary:**

The paper considers networks that abide the Dale's law: a neuron can be either excitatory or inhibitory, but not both simultaneously. In experiments, they show that Daleian networks can approximate non-Daleian ones very well, are more robust to noise and are easier to train using fewer parameters.

**Questions:**

### Main questions
I guess the biggest take away for me: we can use models that violate Dale’s law, because there’s always a similar model that doesn’t. Which brings out two questions:
1) Wouldn’t the Dale law abiding learning algorithms be different? The answer is probably yes, see Clark et al. 2021 (https://proceedings.neurips.cc/paper/2021/hash/532b81fa223a1b1ec74139a5b8151d12-Abstract.html). I think it's worth discussing in more detail.

2) If you take a trained deep network and converted it into a Daleian one (e.g. using a subset of training data, making sure the accuracy didn’t change much), would it produce more brain-like responses (see http://www.brain-score.org/ and their papers)? I’m fine with this being in the discussion, but it’d be great to have it in the results.

The authors should mention why a theoretical analysis of the Daleian approximation is hard (because it’s non-convex, I assume).

Sec. 2.4 is a really interesting result. Maybe it would be possible to run the same experiment for at least CIFAR10 and a small conv net? (Not for the rebuttal deadline, but a comment on that would be appreciated).

### Small corrections
Line 32: “directed” is usually used to describe graphs, rather than networks. Maybe say asymmetric or remove it altogether?

Maybe Eq. 2 can include the definition of D_JS

Lines 167-168: unfinished sentence. …how capable Daleian networks are?

Line 192: move to an equation (not inline)

Line 306: citation needed on log-normal distribution (although I agree it’s widely known)

**Limitations:**

I think the paper adequately addressed the limitations and potential negative impacts.

**Strengths And Weaknesses:**

### Strengths
1. The experimental results are important for theoretical neuroscience.
2. The paper is well-written and easy to follow.

### Weaknesses
1. No theoretical analysis of the Daleian approximation and increased noise robustness of Daleian networks.

### Summary
I give the score 8 (strong accept) as the paper addresses an important questions in theoretical neuroscience, showing previously unknown benefits (at least to my knowledge) of Dale's law.

---

> ### Author Response · Authors · 2022-08-02
> **Response to Reviewer eksj**
>
> > Questions:
>
> > Wouldn’t the Dale law abiding learning algorithms be different? The answer is probably yes, see Clark et al. 2021 (https://proceedings.neurips.cc/paper/2021/hash/532b81fa223a1b1ec74139a5b8151d12-Abstract.html). I think it's worth discussing in more detail.
>
> Indeed, learning that preserves the synaptic signs implies that the “trajectories” in networks’ space or “learning algorithms” would be different. We thank the referee for pointing to this point that we have not discussed in detail, and will add this to the camera-ready version. Moreover, we note that while non-Daleian networks have positive and negative synapses which may exist between any pair of neurons, Daleian networks have in effect 4 types of synapses (E->E,I->I, E->I, and I->E), which suggests potentially more versatile ways to modulate learning - either by having different mechanisms operate at each of the synaptic types, or by allowing for different “credit assignment” to each of these types.
>
> We note however, that we are not confident that our understanding of the question about “learning algorithm” is aligned to that of the referee, and would be happy for more guidance during the “discussion period”.
>
> > If you take a trained deep network and converted it into a Daleian one (e.g. using a subset of training data, making sure the accuracy didn’t change much), would it produce more brain-like responses (see http://www.brain-score.org/ and their papers)? I’m fine with this being in the discussion, but it’d be great to have it in the results.
>
> We naturally agree that the possibility of finding similarities to brain-like responses in Daleian networks is an interesting question. In particular, this could imply that Dale’s rule is the source of such biological features. We have not pursued the idea of directly converting  a trained deep network into a Daleian one, and think this would await future work. We would happily add these ideas to the discussion.
>
> However, following the referee’s comment, we did wonder about the implications of Dale rule on the brain--like responses of the neurons. We will add to the camera-ready version an analysis of the correlation structure between neurons that shows that our Daleian networks show a positively skewed distribution of correlations, unlike non-Daleian, similar to what is observed experimentally.
>
> > The authors should mention why a theoretical analysis of the Daleian approximation is hard (because it’s non-convex, I assume).
>
> Indeed. We will add a section discussing potential theoretical analysis ideas (see answer to referee p2FF).
>
> > Sec. 2.4 is a really interesting result. Maybe it would be possible to run the same experiment for at least CIFAR10 and a small conv net? (Not for the rebuttal deadline, but a comment on that would be appreciated).
>
> We did not manage to do this in time for the rebuttal, but will try to add this to the camera-ready version
>
> > Small corrections
>
> We have addressed all these points in the revised manuscript.

---

> > ### Comment · Reviewer_eksJ · 2022-08-06
> > **Response to the authors**
> >
> > Thank you for the response! I'm pretty happy with the updates/clarifications, so I'm leaving the same score (8) but with increased confidence.
> >
> > >  we are not confident that our understanding of the question about “learning algorithm” is aligned to that of the referee
> >
> > It is! It wasn't necessarily a question, I just though it'd be nice to have some sort of discussion on what Dale's law means for learning. The discussion in your comment fits that idea.

---

> > > ### Author Response · Authors · 2022-08-09
> > > **2nd response to eksj**
> > >
> > > Thank you!
> > > Will add that to the Discussion.

---

### Official Review · Reviewer_p2FF · 2022-07-12

**Rating:** 6
**Confidence:** 3
**Soundness:** 3 good
**Presentation:** 4 excellent
**Contribution:** 3 good

**Summary:**

The paper empirically studies the computational benefits of Dale's law of spiking NNs and rate-based NNs. tivity

**Questions:**

1) line 141, gradient descent doesn't keep the sign constraints of Daleian networks. Did you use the gradient projection?
2) Figure 4b, hard to compare the D and nD distributions. It looks that nD (syn) has a lower accuracy than D (syn), which is inconsistent with Figure 4c?
3) Random Daleian NNs are more "informative", doesn't that depend on stimuli distribution?

**Limitations:**

The authors mentioned limitations in network size and the assumption of weight distribution. It's also worth noticing that there are more excitatory neurons than inhibitory neurons in the cortex, but in this work they are balanced.

**Strengths And Weaknesses:**

Strength:
1) The paper studies an important puzzle in computational neuroscience, why the biological neural nets contain the signs of output weights and what could be the computational implications. The paper is well written and all central claims are well supported by abundant experimental evidence.
2) The functional comparison of random and learned Daleian and non-Daleian NNs is a novel analytical tool to probe the computational consequence of computational constraints.
3) Sensitivity analysis of synaptic weights shows a surprising robustness benefit of Daleian NNs.

Weakness:
Lack of theoretical justifications to better understand
- when does a non-Daleian NN hard to be approximated by a Daleian NN
- why is a Daleian NN less sensitive to weight perturbations
- why does non-Daleian neuronal learning perform worse than Daleian neuronal learning

---

> ### Author Response · Authors · 2022-08-02
> **Response to Reviewer p2FF**
>
> > Weakness: Lack of theoretical justifications to better understand
>
> We agree that a better theoretical understanding of the results presented in the paper would be an important contribution. However, it is not immediately clear to us how to obtain these, as analytical expressions for the stationary distributions are intractable. It is possible that one might find simplified cases in which obtaining an analytical expression is possible, but this would await future work.
>
> However, we plan to expand the discussion in the camera-ready version to refer to two potential components of such an analysis. One such direction would focus on the dynamics of Daleian and non-Daleian ones and explore the eigenvalue structure (along the lines of references 16-18 in the main text) of such collections of networks, and in particular ones that are similar functionally. The other would rely on their functional capacity, and would try to borrow from the recent work of (Amari, Neural Comp 2020) to ask what are the functional distances of these respective classes of networks.
>
> > Questions:
>
> > line 141, gradient descent doesn't keep the sign constraints of Daleian networks. Did you use the gradient projection?
>
> We did not use gradient projection; instead, we conducted gradient descent on the magnitudes of synaptic weights, and the signs (which were pre-determined at initialization) were added when computing the loss of individual networks. We will further elaborate this on the main text, and release the code to reproduce this.
>
> > Figure 4b, hard to compare the D and nD distributions. It looks that nD (syn) has a lower accuracy than D (syn), which is inconsistent with Figure 4c?
>
> We will improve the terminology used in Figure 4: the y-label of 4b should also be D_func (similarly to 4c), and we will explain in the caption that lower D_func means higher approximation accuracy.
>
> > Random Daleian NNs are more "informative", doesn't that depend on stimuli distribution?
>
> The mutual information clearly depends on the stimuli distribution; in the main text, we presented the results for a uniform stimulus distribution, but we have also explored the case for which P(s) was a more structured distribution (e.g., sampled from a Dirichlet distribution with alpha=1); we will add this result for the camera-ready version.
>
> > Limitations:
>
> > The authors mentioned limitations in network size and the assumption of weight distribution. It's also worth noticing that there are more excitatory neurons than inhibitory neurons in the cortex, but in this work they are balanced.
>
> We have now conducted a similar analysis to the one presented in Figure 2e (approximation accuracy for networks of 100 neurons) - using networks in which the ratio between excitatory and inhibitory neurons was 80%/20%, and obtained similar results to the ones we have presented for the 50%/50% case; we will include add this for the camera-ready version.

---

### Official Review · Reviewer_yFoy · 2022-07-15

**Rating:** 6
**Confidence:** 4
**Soundness:** 3 good
**Presentation:** 4 excellent
**Contribution:** 3 good

**Summary:**

Dale's law is the idea that (most) neurons express only a single neurotransmitter, making them excitatory or inhibitory but not both. This paper studies a few models of neural networks constrained to follow Dale's law and contrasts them with networks where the neurons are free to form excitatory and inhibitory connections. The main results are that "Daleian" networks can generally produce nearly the same output as "non-Daleian" ones. They also present better robustness to noise in the weights and can more easily learn distributions of responses with biologically-plausible rules.

**Questions:**

Why did you choose the models that you did?

If you account for weight values (i.e. as real numbers), what can you say about the measure of Daleian networks vs non-Daleian?

What were the distributions of stimuli and responses that you tested and how do those choices influence the performance of Daleian/not networks?

Sec 2.3: Is studying robustness via the Hessian valid if the signs of W are constrained? Some perturbations could break Dale's law.

Sec 2.4: Please explain the task more clearly. How is it similar/different from a machine learning task? Where does the gradient come from and what

Sec 3: Dale's law certainly *has* been studied in the machine learning community before, as recognized in the introduction. However, the discussion reads like this past work didn't occur. Discuss in more detail how your results fit into previous work.

**Limitations:**

The models seem narrow to me, can the authors better motivate them and better discuss their limitations?

The last paragraph of section 3 discusses limitations much too briefly, expand.

**Strengths And Weaknesses:**

Strengths: The paper is very well-written. A couple of places in the margin I wrote "interesting" or "good". This is an important neuro-AI area of study and a significant difference between how networks are modeled by computer scientists and the biological reality. The figures are clear and look nice. Code is available for reproducibility.

Weaknesses:

The study is limited to only certain kinds of network models. Most notably, this work does *not* study artificial neural network models and tasks prevalent in machine learning, so some of the results might be misleading or confusing to the NeurIPS community. In particular, the only models used are a max-entropy style statistical model of recurrent dynamics and a simple dynamical RNN, both linear (in the max-ent model the sufficient statistics are linear) in the previous system state (in contrast to nonlinearities used for machine learning and present in real brains). Furthermore, the stimuli are assumed to be static vectors (eqn 1). The networks don't really have to solve any kind of task, as far as I can tell, except to reproduce certain distributions. These model choices are minimally motivated in the text, despite standing out to me as not standard compared to the work I know.

My confidence in the results is lower because it feels like a lot of details were relegated to the supplement, which I did not have time to read.

Small points by line

1: "implies" seems like wrong word

2: "number of possible architectures" this sentence struck me as imprecise, is this a new (minor) result? If so present as such.

89: "whose entries are all larger than zero" is not obvious, explain

100: $sign(i)$ is incorrect notation, all indices $i > 0$, what you mean is there is a label $s_i \in \pm 1$ for each neuron $i$

106: "100 ms" is arbitrary, you make no connection to a timescale in the text, the only mention of a step size I see is in Fig 1b, which was hard to find

181: Introducing this new model seems to require a new section. I had a hard time distinguishing between models 1 & 2 throughout the results, as the paper switches back and forth. Be explicit throughout.

183: add comma after "Here" in "Here neural activity"

207: "In learning and optimizing their function" is awkward

---

> ### Author Response · Authors · 2022-08-02
> **Response to Reviewer yFoy**
>
> >Weaknesses
>
> It is indeed the case that our focus is on biological neural networks, and we will streamline the abstract to make that clearer. That said, we hope that the results here would motivate further exploration of these ideas in ANNs. We describe below our motivation and justifications for our choices of models and stimuli, and why we believe that they are the appropriate ones for biological neural networks. Still, following this comment (and one from Reviewer eksj), we aim to incorporate an analysis of a simple Daleian ANN in the camera-ready version
>
> >Small points by line
>
> We will address all these points in the camera-ready version
>
>
> > Questions:
>
> > Why did you choose the models that you did?
>
> We will expand our introduction of the models we used and their motivations. In particular, the model from Eq. 1 follows a long history of statistical models used to describe the spiking patterns of neural populations in response to different classes of stimuli. Importantly, log-linear models, Generalized Linear models, Pairwise Maximum Entropy models, and Hidden Markov Model ones, are all variants of the model we have used here for spiking neurons. The dependency between network states is linear in the previous state, but this still implies that the activity of each neuron is given by a sigmoid function over a linear sum of inputs. This form is consistent with experimental observations of stimulus selectivity or receptive field of single neurons (and is the common choice for neurons in ANNs). Moreover, these kinds of models have proved to be highly accurate in capturing the detailed spiking activity patterns of neural populations. We therefore believe this class of models is the appropriate one to explore here, but will better introduce this reasoning in the text.
> We will split the introduction and figures to give the firing rate models of large populations their due place, clarifying the types of models and cases we consider. In particular, we will emphasize that the firing rate model we used is the “standard model” for dynamics of large neural populations in the literature
>
> > If you account for weight values (i.e. as real numbers), what can you say about the measure of Daleian networks vs non-Daleian?
>
> We are not aware of a straightforward way to calculate the measure of the set of Daleian networks with continuous weights. However, one way to generalize the derivation in SI B.1 would be to compute the ratio of the number of Daleian vs. non-Daleian architectures in which the magnitudes of synaptic weights can come from a finite set of numbers (i.e., {0, w_1, w_2,…, w_k} instead of just {0,1}); we would be happy to include such a derivation for the camera-ready version
>
> > What were the distributions of stimuli and responses that you tested and how do those choices influence the performance of Daleian/not networks?
>
> Regarding the stimuli we used -- these are indeed limited compared to the general case of time dependent stimuli, but a step up from many models in the literature that consider a single scalar input to a network. However, since part of our results here are that our networks converge to their steady state distribution of activity relatively fast (which is the one we used to analyze them) - this makes the case of stationary stimuli relevant for the biological case. We hope to pursue other stimuli statistics for the camera-ready version; the case of general time-dependent stimuli would await future manuscripts
> As for the responses - we stress that these are not “chosen”, but are the result of the chosen stimuli parameters, network parameters, and the neuronal dynamics
>
> >Sec 2.3: Is studying robustness via the Hessian valid if the signs of W are constrained?
>
> Since the Hessians are computed using Automatic Differentiation, their computation is exact (unlike finite-differences-based approach, which would indeed violate Dale’s law when the step size > the magnitude of the synapse). Therefore, infinitesimal perturbations do not change the sign of the synapse for non-zero synapses; since the considered networks were fully connected, perturbations did not break Dale’s law
>
> >Sec 2.4: Please explain the task more clearly. How is it similar/different from a machine learning task? Where does the gradient come from and what
>
> For the camera-ready version, we would explicitly write the objective function – the Jensen-Shannon divergence between an empirical neuronal response distribution and the response distribution of Daleian/non-Daleian networks. As before, all gradients are computed using Automatic Differentiation of Python code, using the JAX framework
>
> >Sec 3: Dale's law certainly has been studied in the machine learning community before, as recognized in the introduction. However, the discussion reads like this past work didn't occur. Discuss in more detail how your results fit into previous work
>
> We will expand the Discussion to include a more in-depth review of prior work

---

> > ### Comment · Reviewer_yFoy · 2022-08-08
> > **Evaluations adjusted**
> >
> > I think if the authors were to implement the changes that I proposed, then the paper would be much improved.
> >
> > The one point that I think was missed is that, even if the Hessians are evaluated exactly, any weights that are at the constraint boundary (weights = 0) should not be perturbed even infinitesimally in the direction of negative weight. Have the authors thought of this possibility?
> >
> > I think I will adjust my score slightly upwards with the expectation that the authors will carry out the changes they have said.

---

> > > ### Author Response · Authors · 2022-08-09
> > > **2nd response to yFoy**
> > >
> > > > The one point that I think was missed is that, even if the Hessians are evaluated exactly, any weights that are at the constraint boundary (weights = 0) should not be perturbed even infinitesimally in the direction of negative weight. Have the authors thought of this possibility?
> > >
> > > Indeed, for the case of weight = 0, even infinitesimal perturbations would violate Dale’s law; however, since the networks we considered are fully connected and the synaptic weights are normally distributed, the probability of sampling a synapse whose weight is exactly zero is effectively zero, as well (we have verified this empirically in our code).
> > >
> > > We will add to the final version a sensitivity analysis of sparser networks (for networks of 100 neurons and rate-based dynamics), in which the Hessians would be computed only w.r.t non-zero synapses.
> > >
> > > > I think I will adjust my score slightly upwards with the expectation that the authors will carry out the changes they have said.
> > >
> > > Thank you!
> > >
> > > We are in the midst of implementing the changes proposed by the referees, as described in our initial response, and indeed believe they would make the manuscript significantly better.

---

### Meta-Review · Area_Chair_7ahZ · 2022-08-23

**Recommendation:** Accept
**Confidence:** Certain

**Metareview:**

This paper examines the impact of Dale's principal from neuroscience on neural network computation. Dale's principle says that neurons only release a single neurotransmitter type from their axons, which in principle, means neurons are either excitatory or inhibitory, but not both. Using both spiking and rate-based recurrent networks, the authors show that networks that respect Dale's principle can recapitulate the same computations as those that do not while exhibiting greater robustness to noise. This provides some account for why Dale's principle may provide actual benefits to neural computation.

The reviewers agreed that this paper is well-written and addresses an important question. The decision to accept was unanimous.

**Award:**

No

---

### Decision · Program_Chairs · 2022-09-14

Accept